# Topography of associations between cardiovascular risk factors and myelin loss in the ageing human brain

Olga Trofimova [1,2,3], Adeliya Latypova[1], Giulia DiDomenicantonio [1], Antoine Lutti[1],
Ann-Marie G. de Lange[1,4,5], Matthias Kliegel[6], Silvia Stringhini[7,8,9], Pedro Marques-Vidal [10], Julien Vaucher[10],
Peter Vollenweider[10], Marie-Pierre F. Strippoli [11], Martin Preisig[11], Ferath Kherif [1] & Bogdan Draganski [1,12✉]

Our knowledge of the mechanisms underlying the vulnerability of the brain's white matter microstructure to cardiovascular risk factors (CVRFs) is still limited. We used a quantitative magnetic resonance imaging (MRI) protocol in a single centre setting to investigate the cross-sectional association between CVRFs and brain tissue properties of white matter tracts in a large community-dwelling cohort ($n = 1104$, age range 46–87 years). Arterial hypertension was associated with lower myelin and axonal density MRI indices, paralleled by higher extracellular water content. Obesity showed similar associations, though with myelin difference only in male participants. Associations between CVRFs and white matter microstructure were observed predominantly in limbic and prefrontal tracts. Additional genetic, lifestyle and psychiatric factors did not modulate these results, but moderate-to-vigorous physical activity was linked to higher myelin content independently of CVRFs. Our findings complement previously described CVRF-related changes in brain water diffusion properties pointing towards myelin loss and neuroinflammation rather than neurodegeneration.

[1] Laboratory for Research in Neuroimaging LREN, Centre for Research in Neurosciences, Department of Clinical Neurosciences, Lausanne University Hospital and University of Lausanne, Lausanne, Switzerland. [2] Department of Computational Biology, University of Lausanne, Lausanne, Switzerland. [3] Swiss Institute of Bioinformatics, Lausanne, Switzerland. [4] Department of Psychology, University of Oslo, Oslo, Norway. [5] Department of Psychiatry, University of Oxford, Oxford, UK. [6] Department of Psychology, University of Geneva, Geneva, Switzerland. [7] Center for Primary Care and Public Health (Unisanté), University of Lausanne, Lausanne, Switzerland. [8] Institute of Social and Preventive Medicine, Lausanne University Hospital, Lausanne, Switzerland. [9] Unit of Population Epidemiology, Division of Primary Care Medicine, Geneva University Hospitals and Faculty of Medicine, University of Geneva, Geneva, Switzerland. [10] Department of Medicine, Internal Medicine, Lausanne University Hospital and University of Lausanne, Lausanne, Switzerland. [11] Center for Research in Psychiatric Epidemiology and Psychopathology, Department of Psychiatry, Lausanne University Hospital and University of Lausanne, Lausanne, Switzerland. [12] Neurology Department, Max-Planck-Institute for Human Cognitive and Brain Sciences, Leipzig, Germany. ✉email: bogdan.draganski@chuv.ch

Despite major advances in quantifying the effects of individual cardiovascular risk factors (CVRFs) on brain and behaviour, our knowledge of the underlying neurobiological mechanisms and interactions with genetic, environmental and modifiable lifestyle factors is still limited. Particularly in the context of increased longevity, the question of potentially differential impact of ageing-associated accumulation of CVRFs on the brain's micro-vasculature and parenchyma remains disputed.

Traditionally, CVRFs are most often associated with cerebral large and small vessel disease (cSVD), which determines individuals' negative health outcome[1]. One of the characteristic diagnostic brain imaging correlates of CVRF-related cSVD detectable in magnetic resonance imaging (MRI) is white matter hyperintensity (WMH)[2] load. At the microstructural level WMHs show heterogeneous histological characteristics including varying degrees of demyelination, loss of oligodendrocytes, axonal degeneration, astrogliosis and parenchymal oedema[2,3]. Similarly, the topology of WMH distribution across the brain—periventricular versus subcortical—is linked to differential histological WMH properties and resulting clinical phenotypes[3,4].

The remaining cSVD features—lacunes, enlarged perivascular spaces, cerebral microbleeds, cortical and subcortical micro-infarcts—complete the heterogeneous pathophysiological panoply and explain the marked heterogeneity of clinical manifestations[1,5]. The clinical presentation of cSVD reaching far beyond the established association between WMHs in fronto-subcortical projections, executive and motor dysfunction[4,6,7] suggests the implication of cortical areas involved in language, memory and vision[8]. Arterial hypertension is, together with chronic inflammation, diabetes, smoking and arteriosclerosis, among the CVRFs with strongest empirical evidence for an impact on WMH in cSVD, whilst female sex is an established risk factor for microinfarcts and lacunes (for review see ref. [1]).

Given the steadily growing public health importance of ageing-associated cognitive decline, numerous epidemiological studies on community-dwelling and clinically ascertained cohorts have focused on the impact of ageing on the brain's white matter (WM). However, particularly in large-scale studies, there are shortcomings either with respect to their reliance on self-reported CVRFs[9] or not at all including CVRFs[10,11]. There are also methodological challenges that concern the neurobiological interpretation of results obtained from MRI data: The use of conventional diagnostic fluid-attenuated inversion recovery (FLAIR) MR images as measures of fibre demyelination is debated[2,12,13]. The sensitivity of tensor-based MRI measures of water diffusion to the effects of WM fibre properties on cognitive changes remains unclear[14]. Recent attempts to use T1-weighted/T2-weighted MRI ratio maps to study the effects of, for example, aerobic exercise[15] or apolipoprotein ε4 (ApoE4)[16] are hindered by the unclear relationship between these ratio maps and myelin content in WM and inaccuracies due to hardware imperfections (e.g. spatial inhomogeneity of the radiofrequency transmit field B1+[17]). As a result, the study of ageing-related changes in WM fibres in community-dwelling cohorts remains challenging[18,19].

Current advances in MRI physics allow for neurobiological characterisation of brain tissue properties based on biophysical models[20]. The neurite orientation dispersion and density imaging (NODDI) model using diffusion-weighted imaging (DWI) data provides markers of axonal density and unbound tissue water content along with an estimate of fibre orientation dispersion[21]. Relaxometry-based measures of magnetization transfer saturation (MTsat) are indicative of fibre myelination and are therefore complementary to DWI-based measures (for review see ref. [22]).

In this study, we sought to provide a brain anatomy-focused view, whilst accounting for behavioural phenotypes, ApoE genetic risk and lifestyle factors. We combined DWI-derived NODDI measures with MTsat contrast within subject-specific tracts to model axonal density, extra-cellular free water, tract volume and myelin content in a large-scale community-dwelling midlife and late life cohort acquired at a single centre. We found that CVRFs, particularly hypertension, were associated with lower myelin content and axonal density, and with higher water content in WM tracts of predominantly limbic and prefrontal areas. Contrary to our hypothesis, factors such as depression, low educational level, or lack of physical activity did not exacerbate CVRF-related WM differences. Our finding that male participants showed an obesity-related myelin decrease, while female participants did not, highlights possible morphological or metabolic differences between sexes which impact on brain anatomy should be further investigated.

## Results

**Participants**. Of the 1167 participants of the CoLaus|PsyCoLaus study for which complete MRI data was available, 63 were excluded from the analysis based on quantitative quality control sensitive to head motion in the scanner[23] and a visual inspection of images for gross morphological abnormalities (for details, see Methods and ref. [24]). The demographic and CVRF characteristics of the remaining 1104 participants are described in Table 1. The mean age was 60.1 years (SD = 9.1, range 46–87) and 561 (50.8%) were female. The most frequent CVRF was smoking (56.3% of the sample were current or past smokers), followed by high body mass index (BMI; 55% of the sample had a BMI > 25), high waist-to-hip ratio (WHR; 54% with WHR > 0.9 for males or >0.85 for females), dyslipidemia (37%), hypertension (35.7%) and diabetes (6.5%). The mean aggregate CVRF (aCVRF) score, defined as the sum of the six examined CVRFs (see ref. [9]), was 2.4 (SD = 1.5). On average, males had a higher combined CVRF prevalence, higher educational level, larger intracranial volume, lower prevalence of recent major depression, did less moderate-to-vigorous physical activity (MVPA) and consumed more alcohol than females. Age and ApoE risk did not differ between sexes.

**White matter tract segmentation and quantification**. On diffusion-weighted MR images, we applied TractSeg[25]—an automated deep-learning-based method —to segment the WM of each individual into 31 tracts-of-interest composed of 14 association tracts, 4 projection tracts, 6 limbic tracts and 7 segments of the corpus callosum (see Fig. 1 and Methods for details). We then sampled and averaged MRI-derived maps indicative of myelin content (MTsat), axonal density (intra-cellular volume fraction; ICVF), free water (isotropic volume fraction; ISOVF) and tract volume (number of voxels). Mean and standard deviation values for each tract are shown in Supplementary Data 1. In the following sections, we use interchangeably MRI indices names (MTsat, ICVF, ISOVF) and the brain tissue properties they are indicative of (myelin, axonal density, free water) according to the underlying biophysical model[20], aiming to facilitate reading. We acknowledge that the present results refer to MRI maps which are neither direct nor perfect measures of underlying histological tissue properties.

**Linear and quadratic effects of age**. We observed both linear and quadratic associations between age and WM microstructure characteristics. Across all tracts, the following pattern was observed: with older age and age[2], mean MTsat was lower, mean ICVF was lower except in the fornix where it was higher, mean ISOVF was higher, and volume was lower (detailed regression coefficients and p-values are shown in Supplementary Data 2).

**Table 1 Participant characteristics for the whole sample and for males and females separately.**

|  | Overall (n = 1104) | Males (n = 543, 49.2%) | Females (n = 561, 50.8%) | T/χ² | P |
|---|---|---|---|---|---|
| Age (mean, SD), years | 60.1 (9.1) | 59.8 (9.2) | 60.5 (8.9) | 1.3 | 0.2 |
| Hypertension, % | 35.7 | 41.0 | 30.6 | 12.5 | <0.001 |
| Diabetes, % | 6.5 | 10.2 | 2.9 | 22.6 | <0.001 |
| Dyslipidemia, % | 37 | 45.4 | 28.9 | 31.2 | <0.001 |
| BMI (mean, SD), kg/m² | 26.0 (4.5) | 26.7 (3.8) | 25.3 (5.0) | 4.9 | <0.001 |
| High BMI, % | 55 | 65.5 | 44.9 | 46 | <0.001 |
| WHR (mean, SD) | 0.88 (0.09) | 0.94 (0.07) | 0.83 (0.07) | 24.5 | <0.001 |
| High WHR, % | 54 | 71.0 | 37.7 | 120 | <0.001 |
| Smoking |  |  |  | 10.6 | 0.005 |
| Never, % | 43.7 | 39.0 | 48.2 |  |  |
| Past, % | 36.5 | 40.9 | 32.2 |  |  |
| Current, % | 19.8 | 20.1 | 19.6 |  |  |
| aCVRF (mean, SD) | 2.4 (1.5) | 2.9 (1.5) | 2.0 (1.4) | 10.9 | <0.001 |
| TIV (mean, SD), cm³ | 1437.9 (145.9) | 1527.4 (124.5) | 1351.3 (107.8) | 25.2 | <0.001 |
| Education level |  |  |  | 13.2 | 0.001 |
| Low, % | 47.5 | 44.9 | 50.1 |  |  |
| Middle, % | 27.3 | 25.1 | 29.4 |  |  |
| High, % | 25.1 | 29.9 | 20.5 |  |  |
| APOE risk |  |  |  | 1.5 | 0.5 |
| Low, % | 13.7 | 12.8 | 14.7 |  |  |
| Intermediate, % | 64.5 | 63.9 | 65.0 |  |  |
| High, % | 21.8 | 23.3 | 20.3 |  |  |
| Recent atypical MDD, % | 7.3 | 3.8 | 10.7 | 17.5 | <0.001 |
| Recent melancholic MDD, % | 7.8 | 5.8 | 9.8 | 5.5 | 0.02 |
| MVPA (mean, SD), min/day | 184.3 (84.4) | 169.7 (80.1) | 198.9 (86.2) | 5 | <0.001 |
| Alcohol consumption |  |  |  | 82.6 | <0.001 |
| None, % | 20.7 | 13.6 | 27.2 |  |  |
| <1 unit/day, % | 44.4 | 38.6 | 49.7 |  |  |
| 1–2 units/day, % | 22.2 | 27.6 | 17.1 |  |  |
| >2 units/day, % | 12.8 | 20.2 | 5.9 |  |  |

Two-sided t-tests (continuous variables) and chi-square tests (categorical variables) were used to compare males and females. The associated T/χ² statistics and uncorrected p-values are shown.
*aCVRF* aggregate cardiovascular risk factor, *APOE* apolipoprotein E, *BMI* body mass index, *MDD* major depressive disorder, *MVPA* moderate-to-vigorous physical activity, *SD* standard deviation, *TIV* total intracranial volume, *WHR* waist-to-hip ratio.

**Cardiovascular risk factors and white matter microstructure**. Using linear regression models, we investigated cross-sectional associations between WM microstructure and six CVRFs, adjusting for age, age², sex and total intracranial volume. WM microstructure properties were mainly associated with arterial hypertension, high BMI, high WHR and diabetes (Fig. 2 and Supplementary Data 3). Hypertension was associated with increased free water (ISOVF) in all WM tracts except the corpus callosum (CC) rostrum; with decreased myelin (MTsat) in all tracts except the inferior, middle and superior longitudinal fasciculi (ILF, MLF, SLF I) and posterior CC; with decreased axonal density (ICVF) in the arcuate fasciculus (AF), SLF II and III, cingulum (CG), uncinate fasciculus (UF) and anterior CC, but with increased axonal density in the left fornix (FX).

High BMI correlated with decreased myelin in most association, limbic and callosal tracts; with increased water content in most tracts of the right hemisphere; with increased axonal density in the MLF, FX, cortico-spinal tract (CST) and posterior CC; and with decreased volume in the SLF II and III, CG, and CC midbody and isthmus.

High WHR was associated with decreased myelin in left association tracts except the SLF, bilateral limbic tracts and CC genu; with increased water content in the inferior fronto-occipital fasciculus (IFOF), CST, FX, optic radiation (OR) and posterior CC; with decreased volume in the AF, SLF II and III, CST, CG and CC midbody; and with increased axonal density in the FX.

Diabetes was associated with increased water content in the IFOF and projection tracts; with decreased axonal density in the AF, IFOF, SLF II and III, CG, UF and anterior CC; with decreased volume in the UF, projection and callosal tracts; and with decreased myelin in the IFOF.

Dyslipidemia and smoking showed few associations, mainly in the FX with decreased myelin, but increased water content and axonal density. Dyslipidemia was also associated with decreased axonal density in the left SLF and anterior CC, while smoking showed increased water content in the IFOF and OR.

Standardised β coefficients, which represent the difference in SD units between individuals with and without a given CVRF, ranged from −0.26 to −0.14 for MTsat, 0.13 to 0.46 for ISOVF, −0.38 to 0.29 for ICVF, and −0.29 to −0.09 for volume in significant associations, thus corresponding to small-to-intermediate effect sizes. The relatively larger effect sizes observed for diabetes and decreased axonal density remained similar when adjusting for the other five CVRFs, but they were no longer significant due to larger standard errors, indicating higher uncertainty of these associations (see Supplementary Data 3 and Supplementary Fig. 10). On the other hand, diabetes and increased water content effect sizes were lower when adjusting for other CVRFs, which could be due to shared variance with hypertension.

**Tract clustering**. To test whether there was a spatial pattern in WM tracts' associations with CVRFs, we performed a hierarchical clustering of tracts based on their association patterns with CVRFs. We identified four clusters of tracts (Fig. 3a). The first cluster was composed of the bilateral FX and was characterised by a widespread ICVF association with CVRFs (Fig. 3b). The second cluster was centred on the temporo-parietal region (bilateral ILF, left MLF and

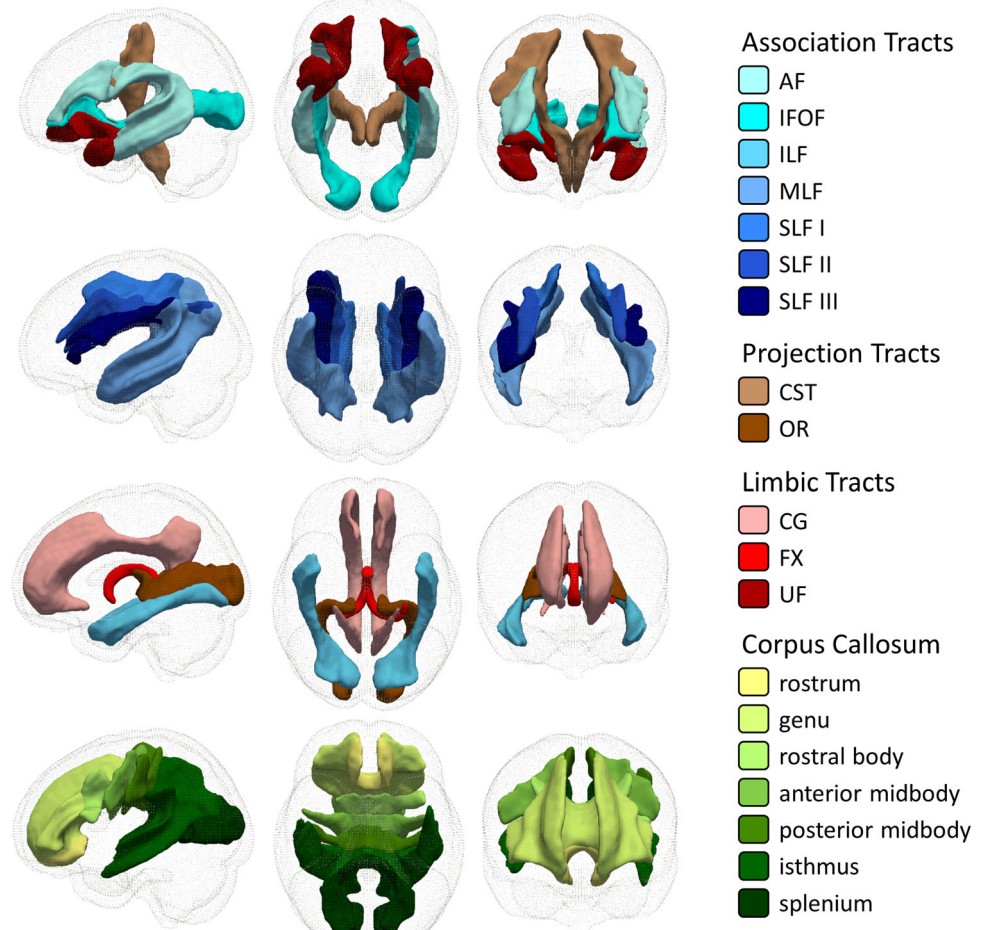

**Fig. 1 Group-average (_n_ = 1104) white matter tracts projected in standard space.** From left to right: sagittal view from the left, axial view from the bottom, coronal view from the front. AF arcuate fasciculus, CG cingulum bundle, CST cortico-spinal tract, FX fornix, IFOF inferior fronto-occipital fasciculus, ILF inferior longitudinal fasciculus, MLF middle longitudinal fasciculus, OR optic radiation, SLF superior longitudinal fasciculus, UF uncinate fasciculus.

right SLF I) and associated with CVRFs exclusively through ISOVF and MTsat. The third cluster contained projection tracts (bilateral CST and right OR) and the posterior CC, to show CVRF associations mainly with ISOVF. The fourth cluster consisted of prefrontal association and limbic tracts (bilateral AF, CG, SLF II and III, and right UF) and the anterior CC. All four MRI maps, especially ICVF and MTsat, determined the association of this cluster with CVRFs. Eight tracts, among which bilateral IFOF and the CC rostrum, did not belong to any cluster based on the defined threshold (65% of the maximal Jaccard distance between tracts). The left UF was close to cluster 2 but did not reach the threshold for inclusion.

**Cumulative cardiovascular risk score**. The aCVRF—a cumulative score consisting of the sum of the six examined CVRFs—correlated positively with water content in almost all tracts; negatively with myelin in all limbic and association tracts, and the anterior CC; negatively with the volume of the AF, IFOF, SLF II, CG, FX and posterior CC; positively with axonal density in the FX and negatively with axonal density in the anterior CC (Fig. 4 and Supplementary Data 4). Standardised βs, corresponding to the SD change in tract parameters with the presence of each additional CVRF, ranged from −0.08 to −0.05 for MTsat, 0.05 to 0.12 for ISOVF, −0.06 to 0.13 for ICVF, and −0.08 to −0.03 for volume.

**Interactions with sex and age**. Interactions between CVRFs and sex were observed mainly for myelin content. Figure 5 shows sex-specific standardised βs for all models where the CVRF × sex interaction was significant. In multiple tracts, MTsat values were negatively associated with high BMI and high WHR in male (β ∈ [−0.40, −0.26]) but not female (β ∈ [−0.06, 0.11]) participants (see Supplementary Data 3 for details). The association between smoking and ISOVF in the AF, SLF I and III, CG, UF and CC genu was positive in male participants (β ∈ [0.23, 0.28]) but negative in female participants (β ∈ [−0.15, −0.05]). In almost all tracts of the right hemisphere, higher aCVRF was linked to lower MTsat in male but not female participants (Fig. 5 and Supplementary Data 4). Full WM microstructure associations with CVRFs are shown for male and female participants separately in Supplementary Figs. 1 and 2.

Age and age² interacted with diabetes in almost all tract volumes. βs associated with both the main effects of age and age² and their interactions with diabetes were negative (Supplementary Data 5), indicating that negative age-related slopes of WM tract volumes were steeper in individuals with diabetes than those without diabetes, as illustrated in Supplementary Fig. 3. The same was true for smoking and volume of the right FX. There were no further significant interactions between CVRFs (including aCVRF) and age or age².

**Seed and target grey matter regions**. For each tract, we defined seed and target cortical grey matter (GM) regions (Fig. 6a), that we analysed for volume and MTsat associations with CVRFs

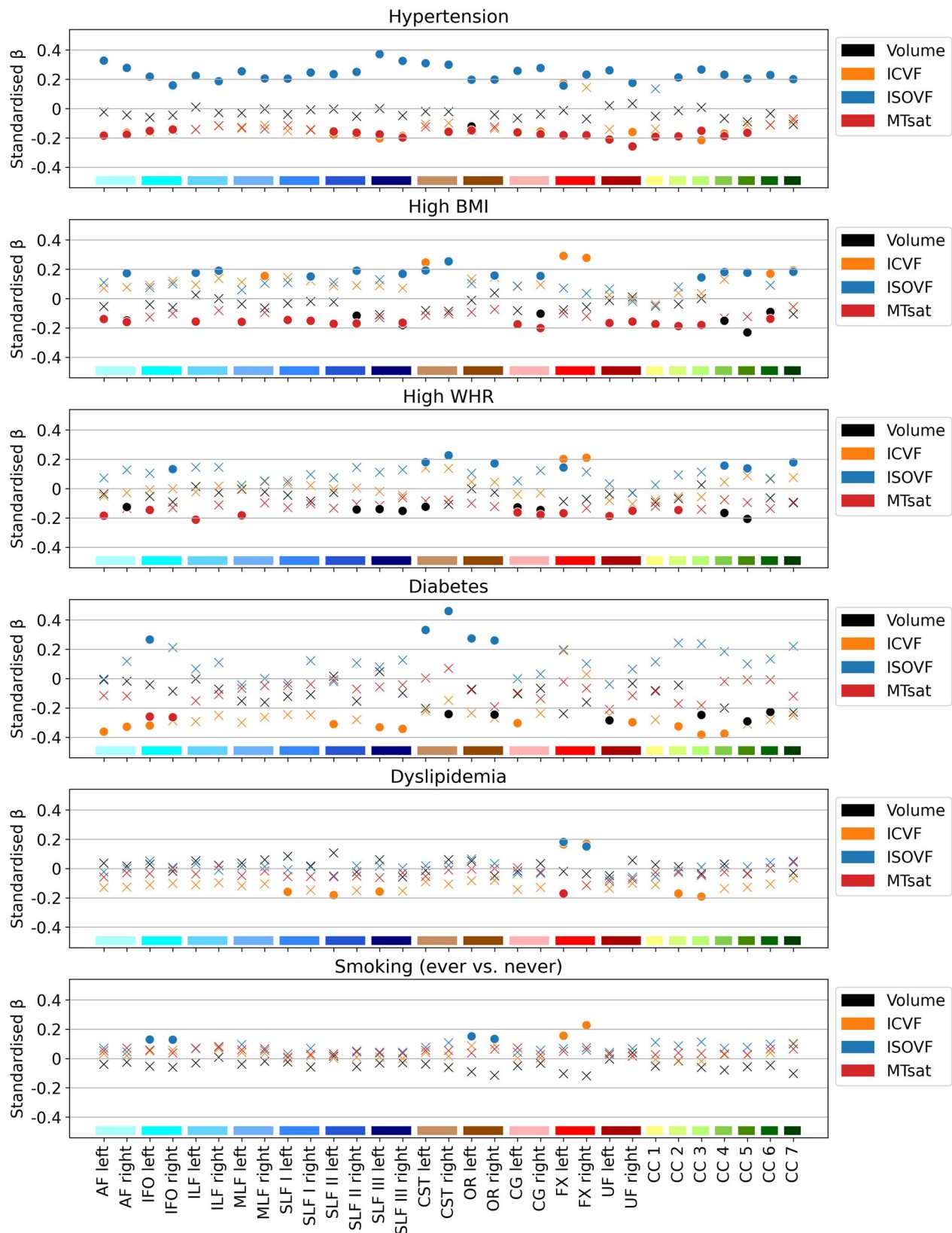

using identical models as for the WM tracts. Figure 6b–d shows standardised βs for seed region, tract and target region associations with arterial hypertension and BMI stratified by sex. WHR stratified by sex, diabetes, dyslipidemia, smoking and aCVRF are shown in Supplementary Fig. 4. Generally, MTsat was more strongly associated with CVRFs than volume. In individuals with

hypertension (Fig. 6b) we observed three patterns: (i) limbic tracts showed lower myelin in the seed structures but not targets, suggesting an anterior-posterior gradient of demyelination; (ii) fronto-parietal (SLF II and III), left fronto-occipital (IFOF) and projection tracts showed a generalised myelin decrease (i.e. in seed, tract and target); and (iii) parietal and occipital GM regions

**Fig. 2 White matter tract microstructure associations with cardiovascular risk factors (n = 1104).** Models were adjusted for age, age$^2$, sex and total intracranial volume. MRI indices of tract-specific axonal density (ICVF), free water (ISOVF), myelin content (MTsat) and tract volume were analysed. Standardised βs are shown as filled circles for significant associations (FDR-corrected p < 0.05) and as crosses for non-significant associations (FDR-corrected p ≥ 0.05). The x-axis contains the 31 tracts-of-interest coloured and grouped, as in Fig. 1, by association tracts (shades of blue), projection tracts (brown), limbic tracts (pink/red) and corpus callosum (yellow/green). AF arcuate fasciculus, BMI body mass index, CC corpus callosum (1 = rostrum, 2 = genu, 3 = rostral body, 4 = anterior midbody, 5 = posterior midbody, 6 = isthmus, 7 = splenium), CG cingulum bundle, CST cortico-spinal tract, FX fornix, ICVF intra-cellular volume fraction, IFO inferior fronto-occipital fasciculus, ILF inferior longitudinal fasciculus, ISOVF isotropic volume fraction, MLF middle longitudinal fasciculus, MTsat magnetization transfer saturation, OR optic radiation, SLF superior longitudinal fasciculus, UF uncinate fasciculus, WHR waist-to-hip ratio.

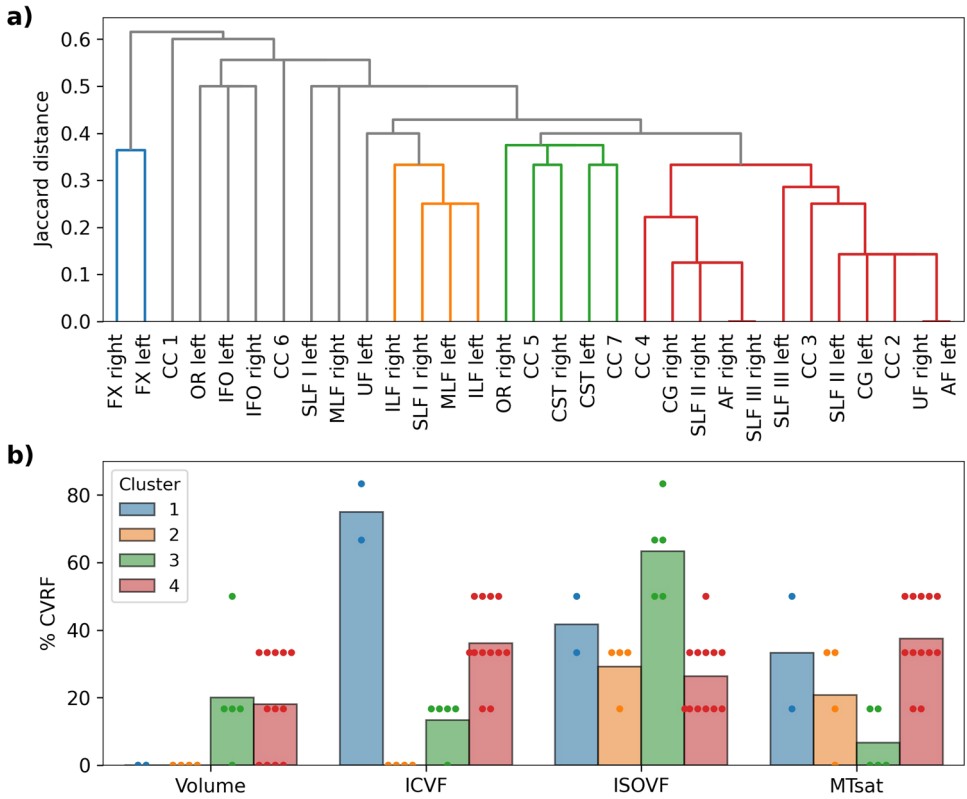

**Fig. 3 Clustering of tracts based on their associations with cardiovascular risk factors. a** Hierarchical clustering of tracts based on the Jaccard distance between patterns of association with CVRFs (significant vs. non-significant). Blue (cluster 1), orange (cluster 2), green (cluster 3) and red (cluster 4) indicate clusters of tracts with inter-tract distance <65% of the maximum distance. Grey indicates all other (i.e., not clustered) tracts. **b** Mean percentage of CVRFs significantly associated with each of the four clusters shown in **a**, for volume, ICVF, ISOVF and MTsat separately. Each dot represents a tract. AF arcuate fasciculus, CC corpus callosum (1 = rostrum, 2 = genu, 3 = rostral body, 4 = anterior midbody, 5 = posterior midbody, 6 = isthmus, 7 = splenium), CG cingulum bundle, CST cortico-spinal tract, FX fornix, ICVF intra-cellular volume fraction, IFO inferior fronto-occipital fasciculus, ILF inferior longitudinal fasciculus, ISOVF isotropic volume fraction, MLF middle longitudinal fasciculus, MTsat magnetization transfer saturation, OR optic radiation, SLF superior longitudinal fasciculus, UF uncinate fasciculus.

— targets of ILF, MLF and SLF I, seeds and targets of CC isthmus and splenium—showed reduced myelin in the absence of significant myelin differences in tracts. In male participants, brain structural correlates of high BMI (Fig. 6c) and WHR (Supplementary Fig. 4a) were largely overlapping. While almost all WM tracts showed lower myelin with higher BMI, this was only the case in a few GM regions, located mainly in the parietal and motor cortices. Volume loss paralleled myelin decrease but with smaller effect sizes. In female participants with high BMI, we observed reduced myelin in the sensorimotor areas. The rest of the brain did not show significant associations with high BMI or WHR, marking a clear contrast with results observed in male participants.

**Models adjusted for education, ApoE risk, recent depression, alcohol consumption and physical activity.** In models that

included, as additional covariates, educational level, ApoE risk, recent major depressive disorder (MDD) with atypical or melancholic episodes, self-reported alcohol consumption and measured MVPA, the sample size was reduced by approximately 45% due to missing data (see Supplementary Data 6 for exact sample sizes). To complement these "adjusted" models, we performed "control" models on the same subset of participant with complete covariate data (i.e. with reduced sample size) but without inclusion of the additional covariates. Compared to previously presented models (with full sample size), control models resulted in similar effect sizes but with many associations not surviving correction for multiple comparisons (see Supplementary Figs. 5 and 7 and Supplementary Data 7), demonstrating the reduced power of the smaller sample to detect effects. The associations previously observed between diabetes and ICVF were no longer present, as a possible result of the small proportion of individuals with diabetes within the sample.

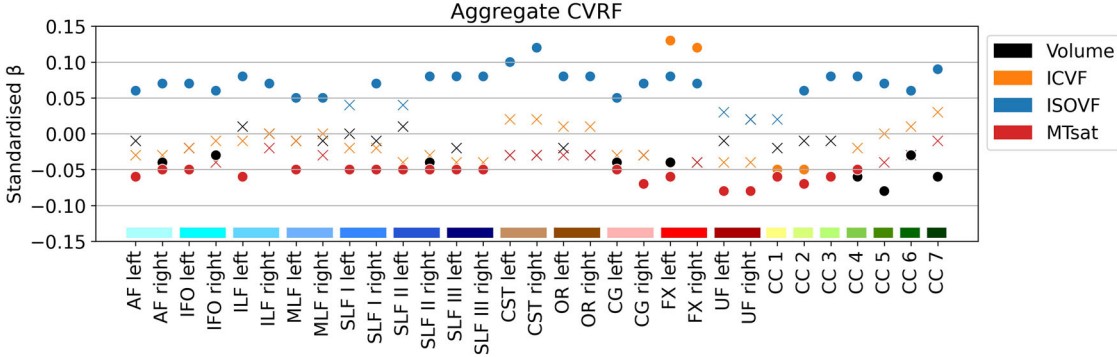

**Fig. 4 White matter tract microstructure associations with the aggregate cardiovascular risk score ($n = 1104$).** Models were adjusted for age, age$^2$, sex and total intracranial volume. MRI indices of tract-specific axonal density (ICVF), free water (ISOVF), myelin content (MTsat) and tract volume were analysed. Standardised βs are shown as filled circles for significant associations (FDR-corrected $p < 0.05$) and as crosses for non-significant associations (FDR-corrected $p \geq 0.05$). The x-axis contains the 31 tracts-of-interest coloured and grouped, as in Fig. 1, by association tracts (shades of blue), projection tracts (brown), limbic tracts (pink/red) and corpus callosum (yellow/green). AF arcuate fasciculus, CC corpus callosum (1 = rostrum, 2 = genu, 3 = rostral body, 4 = anterior midbody, 5 = posterior midbody, 6 = isthmus, 7 = splenium), CG cingulum bundle, CST cortico-spinal tract, CVRF cardiovascular risk factor, FX fornix, ICVF intra-cellular volume fraction, IFO inferior fronto-occipital fasciculus, ILF inferior longitudinal fasciculus, ISOVF isotropic volume fraction, MLF middle longitudinal fasciculus, MTsat magnetization transfer saturation, OR optic radiation, SLF superior longitudinal fasciculus, UF uncinate fasciculus.

In adjusted models (Supplementary Figs. 6 and 7), results were similar to control models except for MTsat associations with arterial hypertension, high BMI and aCVRF which in multiple tracts were no longer significant when including the additional covariates (for details, see Supplementary Data 6). In those models, post-hoc analysis showed that MVPA explained part of the variance in WM MTsat.

We also tested for CVRF x sex interactions in adjusted and control models. For MTsat across the majority of tracts, the BMI × sex and WHR × sex interactions remained significant (Supplementary Figs. 8 and 9 and Supplementary Data 6 and 7), indicating that none of the tested covariates explained the sex differences observed in myelin associations with high BMI and high WHR in the control models. In adjusted models, interactions between sex and aCVRF were significant in almost all tracts bilaterally, mostly for MTsat, but also for volume, ICVF and ISOVF, and showed deleterious association in male but not female participants. Like in control models, there was a diabetes × age interaction in 14 tracts volume. There were no age interactions with the other CVRFs, including aCVRF.

There was no interaction between CVRFs and any of the six additional covariates, indicating that the observed CVRF-related differences in WM microstructure were the same regardless of education level, ApoE risk, recent MDD, physical activity or alcohol consumption (Supplementary Data 8).

**Lifestyle factors and myelin.** To further investigate the observed sex differences in MTsat associations with high BMI and high WHR, we tested the measures in which female participants had healthier indicators than their male counterparts, and which could thus explain the absence of MTsat reduction in female participants with high BMI or WHR. We identified two differential lifestyle factors—MVPA and alcohol consumption (Table 1). In models adjusted for age, age$^2$, sex, total intracranial volume, high BMI and high WHR, MVPA was positively correlated with MTsat in almost all tracts, with effect sizes ranging from 0.08 to 0.12. Alcohol consumption was not associated with MTsat (see details in Supplementary Data 9). There was no MVPA × sex interaction, indicating that higher MVPA was associated with higher myelin content independently of sex. There was no interaction between MVPA and high BMI or high WHR when tested in males and females separately, indicating

that in both sexes, the BMI-MTsat and WHR-MTsat associations were not different across ranges of physical activity (Supplementary Data 8).

## Discussion

Our extensive analysis of CVRF associations with brain anatomy in a monocentric community-dwelling cohort showed sex-dependent relationships predominantly in the white matter microstructure, on the background of less implicated cortical grey matter. The combination of relaxometry- and diffusion-based biophysical models allows for interpreting our findings as CVRF-related myelin decrease paralleled by increased water content rather than neuronal and/or axonal loss. The obtained spatial patterns of tract- and cortical area- specific vulnerability show the complex interactions between individual CVRFs, lifestyle and demographics, integrating previous findings focusing on either white or grey matter micro- and macrostructure.

One of our main findings is the CVRF-related MTsat decrease in WM tracts, beyond and above the effects of age, paralleled by an increase in water content. Corroborating previous findings, arterial hypertension, followed by obesity, was the CVRF with largest overall effect sizes[26–29]. Although MTsat is an indirect measure of myelin content, a recent meta-analysis of MRI myelin biomarkers ranked it among the measures showing highest correlation with histology[30]. Of note is our finding of a positive correlation between the objectively measured levels of physical activity and myelin content in the majority of WM tracts, independent of sex and CVRFs. This is interpreted in the context of previous reports showing that aerobic exercise has beneficial effects on cognition mediated by myelination of late-myelinating WM regions[15,31,32].

Whilst myelin loss can be explained by CVRF-associated arterial stiffness, decreased blood flow and ischemic injury affecting oligodendrocytes[1,33], our findings of increased water content can have multiple origins, including neuroinflammation[34], microinfarct-induced oedema[35,36], or simply cerebro-spinal fluid filling the space left by lost myelin[37]. One of the few studies examining microstructural correlates of hypertension using the NODDI model also found a general increase in ISOVF which the authors interpret as increased free water content resulting from pro-inflammatory immune

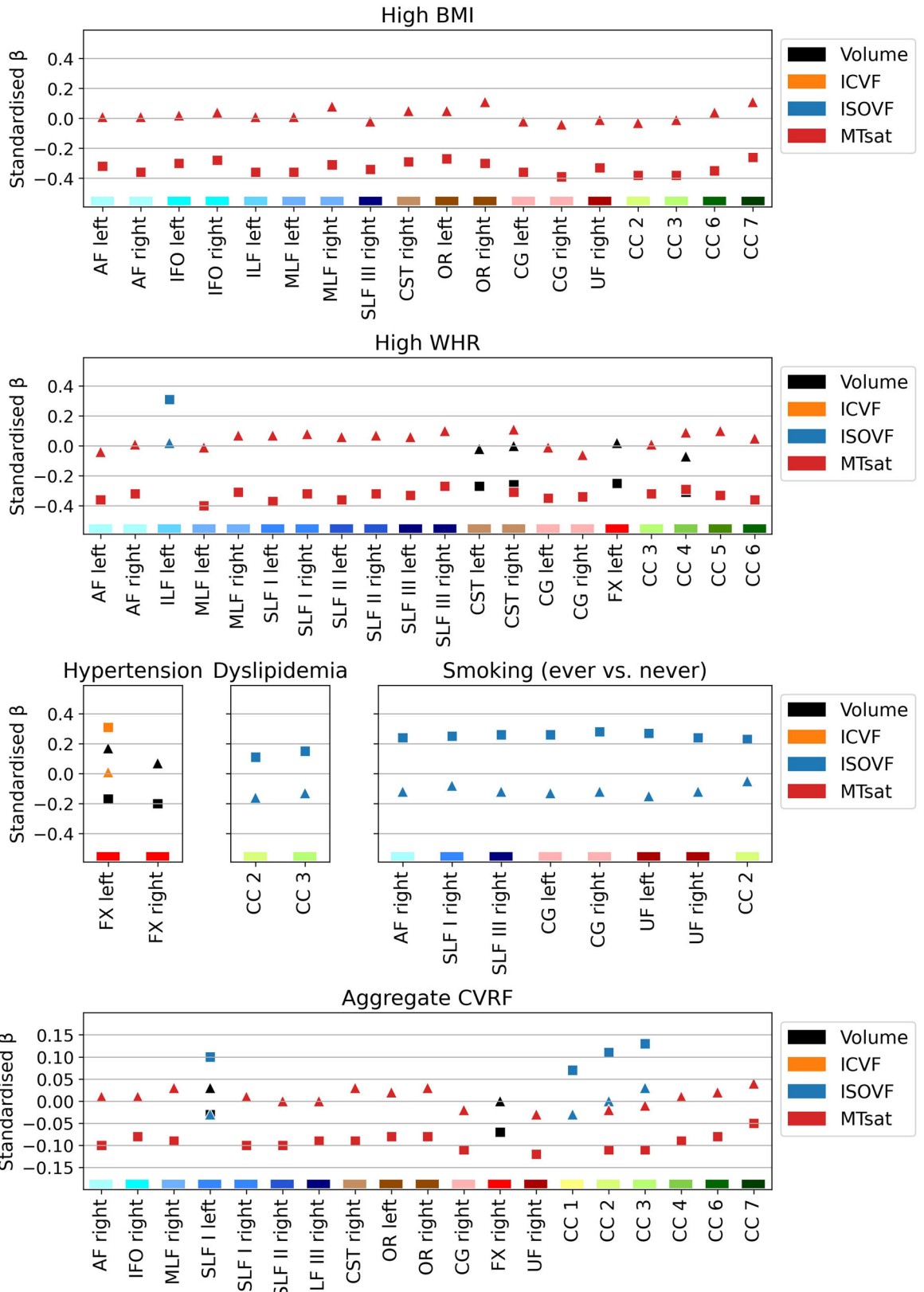

activation and tissue injury, with a possible increase in blood-brain barrier permeability[38].

Finally, tract volume was the brain metric least associated with CVRFs, underscoring the advantages of more specific microstructural measures such as MTsat, ICVF and ISOVF. Indeed, a volume reduction can be interpreted as resulting from

demyelination, axonal loss, loss of glial cells, or a combination of those without the possibility of disentangling the different components.

Using hierarchical clustering, we identified prefrontal and limbic tracts associations characterised by CVRF-related differences in myelin, axonal density, water content and tract volume,

**Fig. 5 For models with significant CVRF × sex interaction, white matter microstructure associations with CVRFs shown separately for males (squares, _n_ = 543) and females (triangles, _n_ = 561).** MRI indices of tract-specific axonal density (ICVF), free water (ISOVF), myelin content (MTsat) and tract volume were analysed. Sex-specific models included age, age[2] and total intracranial volume as covariates. The _x_-axis contains the 31 tracts-of-interest coloured and grouped, as in Fig. 1, by association tracts (shades of blue), projection tracts (brown), limbic tracts (pink/red) and corpus callosum (yellow/green). AF arcuate fasciculus, BMI body mass index, CC corpus callosum (1 = rostrum, 2 = genu, 3 = rostral body, 4 = anterior midbody, 5 = posterior midbody, 6 = isthmus, 7 = splenium), CG cingulum bundle, CST cortico-spinal tract, CVRF cardiovascular risk factor, FX fornix, ICVF intra-cellular volume fraction, IFO inferior fronto-occipital fasciculus, ILF inferior longitudinal fasciculus, ISOVF isotropic volume fraction, MLF middle longitudinal fasciculus, MTsat magnetization transfer saturation, OR optic radiation, SLF superior longitudinal fasciculus, UF uncinate fasciculus, WHR waist-to-hip ratio.

pointing towards increased vulnerability. Supporting the plausibility of our findings, the anterior but not posterior cortical areas adherent to the very same limbic tracts showed reduced intra-cortical myelin, reinforcing the hypothesis of an anterior-posterior gradient of vulnerability[28,39]. At first glance, these results are at odds with the post mortem report about the lack of such a gradient[40]. However, given that our findings are adjusted for the linear and non-linear effects of age, the inference on anterior-posterior gradient of myelin loss is pertinent to the effects of CVRFs rather than to ageing as outlined in the aforementioned study.

The obtained spatial patterns can also be interpreted from the temporal sequence of events viewpoint. Despite the cross-sectional nature of our study, our findings fit into the hypothesis of antero-posterior spread of pathology[41], but also follow the postulated _last-in-first-out_ hypothesis, showing an increased vulnerability of late myelinating prefrontal brain regions[28,42–44] including the cingulum, uncinate fasciculus and superior longitudinal fasciculus which were shown to reach peak myelination later in life[43,45,46]. Likewise, there is cumulating evidence for microvascular damage starting in the WM and spreading to the GM[47]. Given the predominance of MTsat decrease compared to ICVF decrease in our results, we can speculate that CVRF-related axonal loss is secondary to WM demyelination, as evidenced from animal and human studies showing that axons are relatively preserved in demyelinated areas up to later stages of myelin loss[39,48–50].

We report sex differences in the associations between MTsat and high BMI and WHR corroborating previous findings[51–53]. While male participants with high BMI or WHR had less myelin on average than those with normal BMI or WHR, females with high and with normal BMI or WHR had similar levels of myelin. This remained true when adjusting for additional covariates including education, ApoE risk, recent depression and lifestyle factors. The most plausible explanation for this finding lies in presumable sex differences in body fat distribution (subcutaneous vs. visceral abdominal fat), as shown previously[54–56], suggesting that systemic inflammation associated with increased visceral fat could affect myelination of WM tracts[56]. Alternative interpretations include hormonal factors[53,57] or the increased prevalence of obesity-related sleep apnea in males[58,59]. Our results further emphasise the importance of including sex as a factor of interest in studies addressing CVRF and brain ageing[60,61].

Contrary to our hypothesis, none of the additional behavioural and lifestyle factors modified the associations between CVRFs and WM microstructure. Previous studies have reported that MDD subtypes correlated with CVRFs, but with limited impact on brain anatomy[62,63]. The modulating role of the ApoE4 allele in the association between WM lesion and hypertension[64] or dyslipidemia[65] was not replicated in our study. A possible explanation is the broader definition that we adopted for "high" ApoE risk, i.e. carrying at least one ε4 allele, as opposed to ε4/ε4 genotype in the cited studies. Previous research suggested a protective effect of educational attainment against brain ageing and cognitive decline through the concept of cognitive reserve[66–68], but this view is challenged by conflicting results[69,70], among which a recent large-scale longitudinal MRI study that found no protective effect of education[71]. Our results support the latter, with no interaction effects found between age and educational level in predicting variance in WM microstructure. Physical activity explained part of the MTsat variance but did not fully account for the CVRFs-brain associations. Part of the remaining unexplained variance could be explained by genes associated with WM microvascular changes[72–74], although the overall contribution of genetic factors is thought to be small[75].

We acknowledge several limitations of our study. The cross-sectional nature of the study does not allow causal inference about the observed associations. The study sample comes from an urban Swiss population; thus, results might not generalise outside high income countries with predominant Caucasian ethnicity. The relatively large effects reported for diabetes should be interpreted with caution given the sample imbalance (6.5% of participants had diabetes) and the high uncertainty of estimates, especially when adjusting for the other CVRFs. Given the focus on cortico-cortical and cortico-spinal main connections, our analysis did not include striatal, thalamic and cerebellar white matter tracts, which will be included in future studies. Finally, we used tract-averaged brain metrics, so the spatial distribution of effects within tracts remains unknown.

In conclusion, our results provide evidence for a CVRF-related anterior-posterior gradient of vulnerability. Moderate-to-vigorous physical activity was associated with higher myelin content independently of CVRFs, suggesting reversibility of CVRF-related demyelination through aerobic exercise, which should be tested with intervention studies. The combination of MTsat and DWI constitutes an improved radiological marker for early detection of clinically silent cSVD, a risk factor of cognitive decline. Follow-up longitudinal analysis of the CoLaus|PsyCoLaus cohort should investigate the temporality of WM changes and the predictive value of the reported associations on clinically relevant outcomes.

## Methods

**Participants.** Our analysis comprised BrainLaus participants[24]—a nested project within the CoLaus|PsyCoLaus cohort, a prospective study designed to evaluate the links between cardiovascular risk factors and mental health in the general population. Detailed description of the recruitment procedure is available elsewhere[76,77]. In brief, 6734 individuals aged 35 to 75 years were recruited between 2003 and 2006 (baseline) from the civil registry of the city of Lausanne in Switzerland. There were three follow-up evaluations, one from 2009 to 2013 (first follow-up), a second one from 2014 to 2018 (second follow-up) and another one from 2018 to 2022 (third follow-up). During the second follow-up, 1324 participants also took part in the brain magnetic resonance imaging (MRI) investigation (BrainLaus study), among whom 1167 participants completed the full MRI acquisition protocol and 157 interrupted it before the end of the protocol. The CoLaus|PsyCoLaus study received approval from the Ethics Commission of Canton Vaud (www.cer-vd.ch) and participants signed written informed consent prior to inclusion.

**MRI acquisition.** We acquired MRI data on a 3 T whole-body system (Magnetom Prisma - Siemens, Erlangen - Germany), with a 64-channel radiofrequency receive head coil and body coil for transmission. The quantitative MRI protocol included three multi-echo 3D fast low angle shot (FLASH) acquisitions with magnetization transfer-weighted (MTw: TR = 24.5 ms, α = 6°), proton density-weighted (PDw: TR = 24.5 ms, α = 6°) and T1-weighted (T1w: TR = 24.5 ms, α = 21°) contrasts

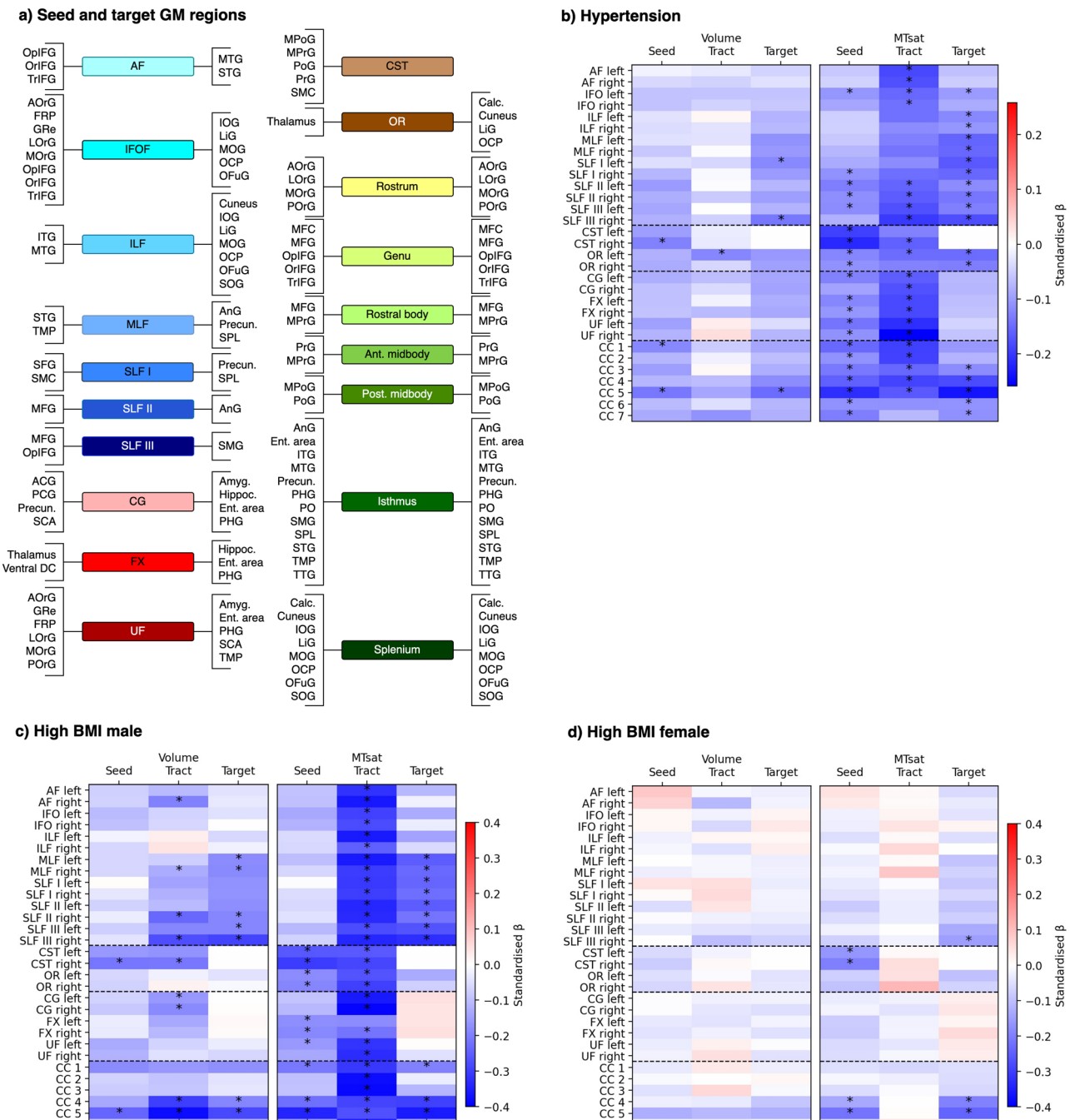

**Fig. 6 CVRF associations with volume and MTsat in white matter tracts and their grey matter seed and target regions ($n = 1104$).** Standardised βs are marked with an asterisk for significant associations (FDR-corrected $p < 0.05$). **a** Constituent sub-regions of the seed and target of each tract. **b** GM and WM associations with hypertension. **c**, **d** GM and WM associations with high BMI in **c** male and **d** female participants. A/PCG anterior/posterior cingulate gyrus, AF arcuate fasciculus, Amyg amygdala, AnG angular gyrus, A/L/M/POrG anterior/lateral/middle/posterior orbital gyrus, BMI body mass index, Calc calcarine cortex, CC corpus callosum (1 = rostrum, 2 = genu, 3 = rostral body, 4 = anterior midbody, 5 = posterior midbody, 6 = isthmus, 7 = splenium), CG cingulum bundle, CST cortico-spinal tract, Ent. area entorhinal area, FRP frontal pole, FX fornix, GM grey matter, GRe gyrus rectus, Hippoc hippocampus, IFO inferior fronto-occipital fasciculus, ILF inferior longitudinal fasciculus, I/M/SOG inferior/middle/superior occipital gyrus, I/M/S/ TTG inferior/middle/superior/transverse temporal gyrus, LiG lingual gyrus, MFC medial frontal cortex, M/SFG middle/superior frontal gyrus, MLF middle longitudinal fasciculus, (M)Po/PrG post/precentral gyrus (medial segment), MTsat magnetization transfer saturation, Op/Or/TrIFG opercular/orbital/ triangular part of inferior frontal gyrus, OCP occipital pole, OFuG occipital fusiform gyrus, OR optic radiation, PHG parahippocampal gyrus, PO parietal operculum, Precun precuneus, SCA subcallosal area, SLF superior longitudinal fasciculus, SMC supplementary motor cortex, SMG supramarginal gyrus, SPL superior parietal lobule, TMP temporal pole, UF uncinate fasciculus, Ventral DC ventral diencephalon, WM white matter.

with 1 mm isotropic resolution[24,43,78]. We used B1 maps computed with the 3D echo-planar spin-echo and stimulated echo images[79,80] (4 mm resolution, TE = 39.06 ms, TR = 500 ms) to correct for the effects of radiofrequency transmit field inhomogeneities[79–82].

The diffusion-weighted imaging (DWI) protocol consisted of a 2D echo-planar sequence with the following parameters: TR = 7400 ms, TE = 69 ms, parallel GRAPPA acceleration factor = 2, FoV = 192 × 212 mm², voxel size = 2 × 2 × 2 mm, matrix size = 96 × 106, 70 axial slices, 118 gradient directions (15 at b = 650 s mm$^{-2}$, 30 at b = 1000 s mm$^{-2}$, 60 at b = 2000 s mm$^{-2}$ and 13 at b = 0 interleaved throughout the acquisition)[43]. We also acquired B0-field maps (2D double-echo FLASH sequence with slice thickness = 2 mm, TR = 1020 ms, TE1/TE2 = 10/12.46 ms, α = 90°, BW = 260 Hz/pixel) to correct for geometric distortions in echo-planar imaging data.

**MRI preprocessing**. Quantitative MRI data were processed in the framework of Statistical Parametric Mapping SPM12 (www.fil.ion.ucl.ac.uk/spm; Wellcome Trust Centre for Neuroimaging, London) using customised MATLAB tools (The Mathworks, Sherborn, MA, USA). We performed a quantitative analysis of image degradation due to head motion using the quality index introduced in ref. [83] and a visual inspection for gross abnormalities. The participants' data that did not meet the criteria outlined in ref. [24] were excluded from subsequent analysis. The MTsat maps were calculated using the MTw, PDw and T1w images averaged across all echoes[81,84]. Using the multi-channel "unified segmentation" approach in SPM12 and enhanced tissue priors[85] we obtained from the MTsat and effective proton density data probabilistic maps of grey matter (GM), white matter (WM) and cerebro-spinal fluid (CSF). We calculated individuals' total intracranial volume (TIV) by summing the GM, WM and CSF volumes. Similarly, for subsequent DWI preprocessing, we created a brain mask consisting of the sum of the three tissue compartments.

DWI data were preprocessed with MRtrix3[86] including denoising[87] and Gibbs ringing artefacts removal[88]. We corrected for eddy current distortions and subject movements with the FSL 5.0 EDDY tool[89]. For echo-planar imaging susceptibility distortion correction, we used the acquired B0 maps with the SPM FieldMap toolbox[90]. Bias field was estimated from the mean b = 0 images and corrected for in all DWI data. We then aligned the preprocessed diffusion maps to the MTsat images using SPM12 rigid body registration.

Within MRtrix3, we estimated tissue-specific (GM, WM and CSF) response functions in 100 randomly selected participants using the msmt_5tt algorithm[91]. This was followed by the creation of a group-average response function which was used to calculate the fibre orientation distribution (FOD) maps in all participants based on the multi-shell multi-tissue constrained spherical deconvolution method[91]. Finally, we normalised the intensity of FODs[92] and extracted FOD peaks[93] for tract segmentation.

**Tract segmentation and microstructure characterisation**. For tract segmentation we used TractSeg, a fast and automatic fully convolutional neural network-based WM segmentation method[25]. We selected 31 from the 72 tracts available in TractSeg, thus excluding cerebellar, thalamic and striatal tracts. The selected tracts (Fig. 1) included association fibres (bilateral arcuate fasciculus, inferior fronto-occipital fasciculus, inferior longitudinal fasciculus, middle longitudinal fasciculus and the three segments of the superior longitudinal fasciculus (I, II, III)), projection fibres (bilateral cortico-spinal tract and optic radiation), limbic tracts (bilateral cingulum bundle, fornix and uncinate fasciculus), and segments of the corpus callosum (rostrum, genu, rostral body/premotor, anterior midbody/primary motor, posterior midbody/primary somatosensory, isthmus and splenium). For tract segmentation we used the FOD peaks obtained with MRtrix3 using the pretrained TractSeg model.

We estimated neurite orientation dispersion and density imaging (NODDI)[21] maps of intra-cellular volume fraction (ICVF) and isotropic volume fraction (ISOVF) from multi-shell diffusion data across all acquired b-values using the AMICO toolbox[94]. We sampled and averaged ICVF, ISOVF and MTsat within individual tracts in participants' native space. Additionally, the number of voxels in each tract was used as a proxy for its volume. To facilitate comparison of associations between WM structure and CVRFs, we standardised all tract-specific values by setting them to a mean of zero and a standard deviation of one.

For each tract, we defined seed and target regions in the GM using SPM12 factorisation-based labelling[95] that provided the grounds for calculation of average MTsat and volume for each labelled cortical region. ICVF and ISOVF were not available in the GM. By arbitrary convention, seed regions were set in the left hemisphere for inter-hemispheric connections and anteriorly for intra-hemispheric connections (Fig. 6a).

**Cardiovascular risk factors**. From the available rich set of measures (for details, see[76]) we defined the CVRFs as follows: (i) arterial hypertension—systolic blood pressure ≥ 140 mm Hg and/or a diastolic blood pressure ≥ 90 mm Hg during the visit and/or the presence of anti-hypertensive drug treatment; (ii) diabetes—fasting plasma glucose ≥ 7.0 mmol/L and/or the presence of oral hypoglycaemic or insulin treatment; (iii) dyslipidemia—high-density lipoprotein (HDL) cholesterol < 1.0 mmol/L and/or triglyceride level ≥ 2.2 mmol/L and/or low-density lipoprotein

(LDL) cholesterol ≥ 4.1 mmol/L or hypolipidemic treatment; (iv) body mass index (BMI) was calculated as weight/height² from measurements done during the visit and high BMI was defined as BMI > 25; (v) waist-to-hip ratio (WHR) was calculated from waist and hip circumferences measured during the visit and high WHR was defined as >0.85 for females or >0.9 for males[96]; and (vi) smoking status was derived from the lifestyle questionnaire that included information on previous and current tobacco smoking and it was dichotomised into ever (i.e. current and/or past) vs. never smoked. To assess the cumulative contribution of CVRFs to WM structure, we calculated aggregate CVRF scores (aCVRF) by summing the presence of hypertension, diabetes, dyslipidemia, high BMI, high WHR and current or past smoking[9].

**Additional variables**. Highest educational attainment was assessed by questionnaire and divided into three levels: mandatory school or apprenticeship (low), high school diploma or upper secondary education (middle), and university degree (high). Genotyping was performed on participants with 4 grandparents of European origins[76]. Apolipoprotein ε (ApoE) risk was defined as low for ε2/ε2 and ε2/ε3 genotypes, intermediate for the ε3/ε3 genotype, and high for carriers of at least one ε4 allele, i.e. for ε2/ε4, ε3/ε4, and ε4/ε4 genotypes.

The CoLaus|PsyCoLaus psychiatric evaluation[77] included the semi-structured Diagnostic Interview for Genetic Studies (DIGS)[97]. Major depressive disorder (MDD) was diagnosed according to the Diagnostic and Statistical Manual of Mental Disorders[98]. MDD was defined as recent if it occurred in the interval since the previous psychiatric evaluation (mean inter-evaluation interval 4.8 ± 2.0 years). For atypical episodes the presence of mood reactivity as well as two of the following four features were required: (i) increased appetite, (ii) hypersomnia, (iii) leaden paralysis, and (iv) interpersonal rejection sensitivity. For melancholic episodes a loss of pleasure or a lack of mood reactivity was required as well as three of the following five symptoms: (i) depression regularly worse in the morning, (ii) early morning awakening, (iii) psychomotor retardation or agitation, (iv) decreased appetite, and (v) excessive guilt.

Physical activity was measured with a wrist-worn triaxial accelerometer (GENEActiv, Activinsights Ltd., United Kingdom) for 14 consecutive days[99]. We averaged the daily time in minutes spent in moderate-to-vigorous physical activity across all days with at least 10 h of diurnal wear-time, only when there were at least 5 such weekdays and 2 weekend days. Weekly alcohol consumption was assessed by self-reported questionnaire and grouped into four categories: non-drinkers, low (1–6 units/week), moderate (7–13 units/week), and high (≥14 units/week)[100].

**Statistics and reproducibility**. Analyses were performed on data not exceeding 4 standard deviations (SD) from the sample mean for each tract and map separately (Supplementary Data 1 shows the number of observations meeting this criterion for each tract and map). First, we tested linear and quadratic age effects by ordinary least squares (OLS) linear regressions with equations:

$$Tract\ value = \beta_0 + \beta_1 age + \beta_2 sex + \beta_3 TIV + \varepsilon \tag{1}$$

$$Tract\ value = \beta_0 + \beta_1 age^2 + \beta_2 sex + \beta_3 TIV + \varepsilon \tag{2}$$

where tract value was one of the four MRI maps—mean MTsat, mean ICVF, mean ISOVF or volume—in one of the 31 tracts-of-interest. $\beta_1$ was the estimate of interest.

We then performed OLS regressions with equation:

$$Tract\ value = \beta_0 + \beta_1 CVRF + \beta_2 age + \beta_3 age^2 + \beta_4 sex + \beta_5 TIV + \varepsilon \tag{3}$$

where CVRF was one of the six cardiovascular risk factors—hypertension, diabetes, dyslipidemia, high BMI, high WHR or smoking—and tract value was one of the four maps in one of the 31 tracts. $\beta_1$ was the estimate of interest. Thus, we performed 744 tests in this analysis (6 CVRFs × 4 maps × 31 tracts). CVRFs took value of 0 (absence) or 1 (presence). We then performed a hierarchical clustering of the 31 tracts based on the Jaccard distance between their patterns of association with CVRFs (1 for significant and 0 for non-significant associations). We defined clusters as groups of tracts with distance < 65% of the maximum distance between tracts.

We assessed potential interactions between CVRFs and sex or age with the following models:

$$Tract\ value = \beta_0 + \beta_1 CVRF + \beta_2 age + \beta_3 age^2 + \beta_4 sex + \beta_5 TIV + \beta_6 CVRF*sex + \varepsilon \tag{4}$$

$$Tract\ value = \beta_0 + \beta_1 CVRF + \beta_2 age + \beta_3 age^2 + \beta_4 sex + \beta_5 TIV + \beta_6 CVRF*age + \varepsilon \tag{5}$$

where the interaction term $\beta_6$ was the estimate of interest. In models with significant sex interaction, we computed the same analysis (Eq. 3 without sex) in males and females separately.

To test the cumulative contribution of CVRFs to WM microstructure, we performed the same steps (3–5 but without tract clustering) with the aCVRF score. We then repeated the same analysis (3–5 on individual CVRFs and aCVRF) with six additional covariates—education level, ApoE risk, recent atypical and melancholic MDD, MVPA and alcohol consumption. Since the sample size was substantially reduced due to missing data on the covariates, we also performed 3 to

5 without the additional covariates, but on the subset of participants with complete covariate data, to allow comparison between model results with and without additional covariates.

Furthermore, we tested interaction effects between the additional covariates and CVRFs:

$$Tract\ value = \beta_0 + \beta_1 CVRF + \beta_2 age + \beta_3 age^2 + \beta_4 sex + \beta_5 TIV \\ + \beta_6 covariate + \beta_7 CVRF * covariate + \varepsilon \tag{6}$$

where CVRF was one of the six CVRFs or aCVRF, covariate was one of the six additional covariates, and $\beta_7$ the estimate of interest.

For GM seed and target region analysis, we performed step 3 replacing the tract estimates by GM regional values as outcome, iteratively with the six CVRFs and aCVRF. Given the sex interactions found with BMI and WHR in the WM, we separated male and female participants for analyses of those two CVRFs.

As a post-hoc analysis to sex interactions with high BMI and WHR in predicting MTsat values, we investigated a possible confounding role of lifestyle factors—MVPA and alcohol consumption—and their interaction with sex:

$$Mean\ MTsat = \beta_0 + \beta_1 lifestyle + \beta_2 age + \beta_3 age^2 + \beta_4 sex + \beta_5 TIV \\ + \beta_6 high\ BMI + \beta_7 high\ WHR + \varepsilon \tag{7}$$

$$Mean\ MTsat = \beta_0 + \beta_1 lifestyle + \beta_2 age + \beta_3 age^2 + \beta_4 sex + \beta_5 TIV \\ + \beta_6 high\ BMI + \beta_7 high\ WHR + \beta_8 lifestyle * sex + \varepsilon \tag{8}$$

where lifestyle was MVPA or alcohol consumption and $\beta_1(7)$ and $\beta_8(8)$ the estimates of interest.

We also tested the model described in Eq. 6 separately in males and females where the CVRF was high BMI or high WHR and the covariate was MVPA.

As a post hoc analysis to the relatively large effect sizes observed in diabetes associations, we investigated those associations in models adjusted for all other CVRFs:

$$Tract\ value = \beta_0 + \beta_1 diabetes + \beta_2 hypertension + \beta_3 highBMI + \beta_4 highWHR \\ + \beta_5 dyslip + \beta_6 smoking + \beta_7 age + \beta_8 age^2 + \beta_9 sex + \beta_{10} TIV + \varepsilon \tag{9}$$

where $\beta_1$ was the primary estimate of interest, but $\beta_{2-6}$ are also reported for comparison.

For all analyses, we corrected for multiple comparisons across tracts, maps and CVRFs using false discovery rate (FDR) correction[101]. Alpha threshold was set at .05 on FDR-corrected $p$-values.

**Reporting summary**. Further information on research design is available in the Nature Portfolio Reporting Summary linked to this article.

## Data availability

The CoLaus|PsyCoLaus cohort data used in this study cannot be fully shared as they contain potentially sensitive patient information. As discussed with the competent authority, the Research Ethic Committee of the Canton of Vaud, transferring or directly sharing this data would be a violation of the Swiss legislation aiming to protect the personal rights of participants. Non-identifiable, individual-level data are available for interested researchers, who meet the criteria for access to confidential data sharing, from the CoLaus Datacenter (CHUV, Lausanne, Switzerland). Instructions for gaining access to the CoLaus data used in this study are available at https://www.colaus-psycolaus.ch/professionals/how-to-collaborate/.

## Code availability

The code used for the statistical analysis is available on GitLab: https://gitlab.com/otrofimo/brainlaus_cvrf_wm. The analysis was performed using python 3.8.8 packages pandas 1.2.4, numpy 1.20.1, statsmodels 0.12.2 and scipy 1.6.2.

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

## Acknowledgements

B.D. is supported by the Swiss National Science Foundation (project grants Nr. 32003B_212466, 324730_192755 and CRSII5-3_209510), ERA_NET iSEE project (32NE30_198552), the Swiss Personalized Health Network SACR project, the CLIMACT starting grant UNIL-EPFL, and the Leenaards Foundation. LREN is very grateful to the Roger De Spoelberch and Partridge Foundations for their generous financial support. A.L. is supported by the Swiss National Science Foundation (grant 320030_184784) and the ROGER DE SPOELBERCH Foundation. A.-M.G.d.L. is supported by the Swiss National Science Foundation (grant PZ00P3_193658). The CoLaus|PsyCoLaus study was and is supported by research grants from GlaxoSmithKline, the Faculty of Biology and Medicine of Lausanne, the Swiss National Science Foundation (grants 3200B0–105993, 3200B0-118308, 33CSCO-122661, 33CS30-139468, 33CS30-148401, 33CS30_177535 and 3247730_204523) and the Swiss Personalized Health Network (project: Swiss Ageing Citizen Reference).

## Author contributions

Conceptualisation by O.T., M.K., S.S. and B.D. Study design and methodology by O.T., A.-M.G.d.L., F.K. and B.D. MRI data acquisition by G.D.D., A. Lutti, F.K. and B.D. MRI data preprocessing by O.T., A. Latypova, G.D.D., A. Lutti and B.D. Other data pre-processing by P.M.V., J.V., P.V., M.P.F.S. and M.P. Statistical analysis by O.T. Data visualisation by O.T. and B.D. Funding acquisition by M.K., S.S. and B.D. All authors discussed the results and contributed to the writing and revision of the manuscript.

## Funding

## Competing interests

The authors declare no competing interests.
