## [Peer Review File · Communications Biology]

Topography of associations between cardiovascular risk factors and myelin loss in the ageing human brainReviewers' comments:

Reviewer #1 (Remarks to the Author):

Trofimova et al. investigated the relationship between cardiovascular risk factors and the brain's white matter microstructure using quantitative MRI in a cross-sectional cohort over 1000 participants. The authors identified lower myelin and axonal density together with higher extracellular water content in persons with arterial hypertension and obesity, particularly in male participants. Especially frontal and limbic tracts were affected by these changes. Of interest, higher physical activity was linked to higher myelin content independent of cardiovascular risk factors.

This is a timely and very well written study in a large sample. The methods implemented are state of the art and well described.

There are a few suggestions the authors should consider:

The manuscript is of potential interest to clinicians. The figures are difficult to read. I would recommend to use less abbreviations in the figures. For example, Fig.2, the labels on the right side could be fully mentioned instead of abbreviated.

What is the rationale to investigate non-linear age effects (age²)? Is there evidence for non-linear effects with respect to metabolic diseases?

Why were cerebellar, thalamic and striatal tracts excluded? I think this is unfortunate as these structures are strongly influenced by BMI.

There are previous studies using quantitative MRI in obesity (in smaller sample sizes). These need to be mentioned and discussed. Do the findings of the large cohort confirm these studies or not?

(Example: Metzler-Baddeley 2019, Neuroimage; Kullmann, Callaghan et al. 2016 Neuroimage)

I think the sex specific findings are particularly interesting. Is there a way to check for interactions with specific lifestyle factors? Since physical activity seems to have strong protective effects, it would be nice to know whether women who exercise more regularly show less associations with BMI and cardiovascular factors.

Reviewer #3 (Remarks to the Author):

This manuscript describes cross-sectional brain and particularly brain white matter correlates of cardiovascular risk factors in a population-based study. A novel element of the study is the inclusion of MTsat imaging, which may correlate more closely with myelin content than other measures. The results highlight apparent differences with CVRF in frontal tracts, which the authors relate to those latest to myelinate.

The manuscript is well presented, based on established techniques. The claim the MTsat is a measure of myelin (implied in the results) is overstated, as more carefully described in the discussion. I would urge the descriptions of this (and the diffusion derived measures) in the results be related to the imaging sequence/analytical outcome used rather than as the preferred interpretation (i.e., "myelin", "free water" etc). None of these measures can be thought of as a perfect surrogate.

The novelty of the observations is modest, but the clarity of the MTsat data strengthens previous less well founded conclusions. These are worth making known to a larger readership. The question of white matter pathology is highly topical with increasing attention being paid to age-related changes in oligodendroglia and the importance of maintaining robust connectivity. That said, their impact is limited by the bare description in the context of correlational models as beta values without functional reference. Can the authors provide even correlative performance measures of cognition to highlight the magnitude of associations between, e.g., MTsat decreases and cognitive impairment? Even better would be to describe results of mediation analysis if they demonstrated that differences of the

magnitude seen here mediate differences in relationships between CVRF load and clinically meaningful cognitive performance differences. These latter two additional analyses would greatly strengthen the report.

The "last in/first out" hypothesis is rather broadly framed. Can this be more quantitatively framed on the basis of prior imaging studies of regional myelin-sensitive imaging measure changes? This might be by, e.g., tract specific timing of last myelin changes with respect to degree of differences seen here.

Revised version of the manuscript **COMMSBIO-22-3543-T**, “*Neurobiology of associations between cardiovascular risk factors and anatomy of the ageing brain*” by Trofimova et al.

Reviewers' comments:

Reviewer #1 (Remarks to the Author):

Trofimova et al. investigated the relationship between cardiovascular risk factors and the brain's white matter microstructure using quantitative MRI in a cross-sectional cohort over 1000 participants.

The authors identified lower myelin and axonal density together with higher extracellular water content in persons with arterial hypertension and obesity, particularly in male participants. Especially frontal and limbic tracts were affected by these changes. Of interest, higher physical activity was linked to higher myelin content independent of cardiovascular risk factors.

This is a timely and very well written study in a large sample. The methods implemented are state of the art and well described.

We are thankful to the Reviewer for the appreciation of our work.

There are a few suggestions the authors should consider:

- 1. The manuscript is of potential interest to clinicians. The figures are difficult to read. I would recommend to use less abbreviations in the figures. For example, Fig.2, the labels on the right side could be fully mentioned instead of abbreviated.*

Given that five of the co-authors including the senior author are clinicians from different disciplines – psychiatry, general medicine, and neurology, we considered by default avoiding methodology specific abbreviations. In our opinion, the fully spelled out abbreviations in figures – e.g. "Isotropic volume fraction" instead of "ISOVF" or "Magnetization transfer saturation" instead of "MTsat" would not make a big difference to the inexperienced reader. Instead, we expanded the figure legends including both spelled out versions of the abbreviations and their interpretation in neurobiological terms. The authors of a recent publication on brain anatomy differences in children with Developmental Language Disorders using the relaxometry MRI method followed the same strategy¹.

Fig. 2:

"White matter tract microstructure associations with cardiovascular risk factors, adjusted for age, age², sex and total intracranial volume. MRI indices of tract-specific axonal density (ICVF), free water (ISOVF), myelin content (MTsat), and tract volume were analysed."

Fig. 4:

"White matter tract microstructure associations with aCVRF, adjusted for age, age², sex and total intracranial volume. MRI indices of tract-specific axonal density (ICVF), free water (ISOVF), myelin content (MTsat), and tract volume were analysed."

Fig. 5:

"For models where the CVRF x sex interaction was significant (FDR-corrected $p < 0.05$), standardised β s of the association between white matter microstructure and CVRF are shown for male (squares) and female (triangles) participants separately. MRI indices of tract-specific axonal density (ICVF), free water (ISOVF), myelin content (MTsat), and tract volume were analysed."

2. *What is the rationale to investigate non-linear age effects (age²)? Is there evidence for non-linear effects with respect to metabolic diseases?*

When looking at brain anatomy as outcome variable, age is one of the main explanatory variables, i.e. often the regressor with largest effect size, indicating that careful adjustment for age is necessary to avoid false positives. The non-linear associations between age and white matter microstructure are well-established²⁻⁵. Therefore, we include both age and age squared as covariates in our models, aiming to adjust for linear and non-linear age effects.

3. *Why were cerebellar, thalamic and striatal tracts excluded? I think this is unfortunate as these structures are strongly influenced by BMI.*

We agree with the Reviewer that cerebellar, thalamic, and striatal tracts are of potential interest when studying the associations between cardiovascular risk factors and the brain's white matter. The tract segmentation tool we used (TractSeg) includes 72 white matter tracts. Given that we test for the effects of 6 cardiovascular risk factors, their interactions with sex, age and 6 additional factors, plus seed and target grey matter areas for each analysed tract, we decided to focus on tracts that were previously associated with CVRFs. We are currently working on a more parsimonious model that will allow to address this question.

4. *There are previous studies using quantitative MRI in obesity (in smaller sample sizes). These need to be mentioned and discussed. Do the findings of the large cohort confirm these studies or not? (Example: Metzler-Baddeley 2019, Neuroimage; Kullmann, Callaghan et al. 2016 Neuroimage)*

We thank the Reviewer for pointing to these articles affirming the findings and interpretations in our study. We follow the Reviewer's suggestion and integrate these two references in the Discussion:

"Corroborating previous findings, arterial hypertension, followed by obesity, was the CVRF with largest effect size^{27-29, 30}."

"The most plausible explanation for this finding lies in presumable sex differences in body fat distribution (subcutaneous vs. visceral abdominal fat), as shown previously⁵⁵⁻⁵⁷, suggesting that systemic inflammation associated with increased visceral fat could affect myelination of WM tracts⁵⁷."

5. *I think the sex specific findings are particular interesting. Is there a way to check for interactions with specific lifestyle factors? Since physical activity seems to have strong protective effects, it would be nice to know whether women who exercise more regularly show less associations with BMI and cardiovascular factors.*

We agree with the Reviewer that this finding is of interest given its potential implications for preventive action through lifestyle changes. In the whole cohort, we tested whether cardiovascular risk factors (including BMI) interacted with other covariates (including physical activity) with the following model:

$$\text{Tract value} = \beta_0 + \beta_1 \text{CVRF} + \beta_2 \text{age} + \beta_3 \text{age}^2 + \beta_4 \text{sex} + \beta_5 \text{TIV} + \beta_6 \text{covariate} + \beta_7 \text{CVRF} \\ * \text{covariate} + \varepsilon$$

As reported in the manuscript, the interaction term β_7 was not significant in all tested models, indicating that, for example, there was no difference in the CVRF-brain associations between individuals who did a lot of physical activity and those who did little. We did not test for this interaction term separately among females or males. Following the Reviewer's suggestion, we tested the above model separately in males and females, with high BMI and high WHR as "CVRF" and moderate-to-vigorous physical activity as "covariate". Results were added to Table S7 as two

additional columns (MVPA x CVRF Male and MVPA x CVRF Female). None of the tested interaction terms were significant, suggesting that presumed protective effects of physical activity and deleterious effects of high BMI or WHR were independent and additive rather than multiplicative. The range of p-values for MTsat interactions was 0.14 - 1 for females and 0.07 - 0.98 for males. We included this additional information in the Methods section:

We also tested the model described in equation 6 separately in males and females where the CVRF was high BMI or high WHR and the covariate was MVPA.

Results section (last paragraph):

There was no interaction between MVPA and high BMI or high WHR when tested in males and females separately, indicating that in both sexes, the BMI-MTsat and WHR-MTsat associations were not different across ranges of physical activity (Table S7).

References:

1. Krishnan, S. *et al.* Quantitative MRI reveals differences in striatal myelin in children with DLD. *eLife* 11, e74242 (2022).
2. Slater, D. A. *et al.* Evolution of white matter tract microstructure across the life span. *Human Brain Mapping* (2019) doi:10.1002/hbm.24522.
3. Bouhrara, M. *et al.* Age-related estimates of aggregate *g* -ratio of white matter structures assessed using quantitative magnetic resonance neuroimaging. *Hum Brain Mapp* 42, 2362–2373 (2021).
4. Merenstein, J. L., Corrada, M. M., Kawas, C. H. & Bennett, I. J. Age affects white matter microstructure and episodic memory across the older adult lifespan. *Neurobiology of Aging* 106, 282–291 (2021).
5. Yeatman, J. D., Wandell, B. A. & Mezer, A. A. Lifespan maturation and degeneration of human brain white matter. *Nat Commun* 5, 4932 (2014).

Reviewer #3 (Remarks to the Author):

This manuscript describes cross-sectional brain and particularly brain white matter correlates of cardiovascular risk factors in a population-based study. A novel element of the study is the inclusion of MTsat imaging, which may correlate more closely with myelin content than other measures. The results highlight apparent differences with CVRF in frontal tracts, which the authors relate to those latest to myelinate.

The manuscript is well presented, based on established techniques.

We are thankful to the Reviewer for the correct summary and the appreciation of the novel elements of the study.

- 1. The claim the MTsat is a measure of myelin (implied in the results) is overstated, as more carefully described in the discussion. I would urge the descriptions of this (and the diffusion derived measures) in the results be related to the imaging sequence/analytical outcome used rather than as the preferred interpretation (i.e., "myelin", "free water" etc). None of these measures can be thought of as a perfect surrogate.*

We fully agree with the Reviewer that reporting findings as "myelin" or "free water" is a shortcut and could wrongly suggest that we used a perfect or direct measure of these brain tissue components. As already mentioned by Reviewer#1, the MRI-physics specific terminology is reducing the article's appeal for the broader readership of the journal. Our decision to use those terms was based on the wish to facilitate reading of acronyms such as ICVF or MTsat to non-experts in neuroimaging. The fact is that the community adopted this simplification and recent articles feature qMRI and myelin already in their title: "Quantitative MRI reveals differences in striatal myelin in children with DLD" – Krishnan et al., 2022.

To address the Reviewer's comment, we added the following sentence in the Results, "White matter tract segmentation and quantification" section, to make it clear from the start that the MRI-derived measures we describe are indirect and imperfect measures of the underlying brain tissue:

"In the following sections, we use interchangeably MRI indices names (MTsat, ICVF, ISOVF) and the brain tissue properties they are indicative of (myelin, axonal density, free water) according to the

underlying biophysical model²⁶, aiming to facilitate reading. We acknowledge that the present results refer to MRI maps which are neither direct nor perfect measures of underlying histological tissue properties."

2. *The novelty of the observations is modest, but the clarity of the MTsat data strengthens previous less well founded conclusions. These are worth making known to a larger readership. The question of white matter pathology is highly topical with increasing attention being paid to age-related changes in oligodendroglia and the importance of maintaining robust connectivity. That said, their impact is limited by the bare description in the context of correlational models as beta values without functional reference. Can the authors provide even correlative performance measures of cognition to highlight the magnitude of associations between, e.g., MTsat decreases and cognitive impairment? Even better would be to describe results of mediation analysis if they demonstrated that differences of the magnitude seen here mediate differences in relationships between CVRF load and clinically meaningful cognitive performance differences. These latter two additional analyses would greatly strengthen the report.*

At the current phase, cognitive testing was performed only in subjects older than 65 years, which represented ca. 300 individuals. At that age, the probability of having an additional undiagnosed age-associated disorder – neurodegenerative disease, small vessel disease, malignancy, is much higher than at a younger age and this would represent a major confounding factor for the intended research question. We are currently acquiring “clinically meaningful” cognitive measures in a sample with a larger age range, and will aim at answering the relevant questions as soon as we have completed this stage.

3. *The “last in/first out” hypothesis is rather broadly framed. Can this be more quantitatively framed on the basis of prior imaging studies of regional myelin-sensitive imaging measure changes? This might be by, e.g., tract specific timing of last myelin changes with respect to degree of differences seen here.*

The reference to the “last in/first out” hypothesis is based on our previous study (Slater et al. 2019) where this question was explicitly tested. We agree that the link to the hypothesis would be stronger with a quantitative comparison between, for example, myelin maturation peak age and the degree of CVRF-related MTsat change tract by tract. This is however out of the scope of the

current investigation, since myelin maturation peak is typically derived from datasets that include younger adults, while our cohort starts at 46 years old, i.e. after myelin peak. We can only qualitatively compare our findings to previous studies on myelination trajectories. According to the Reviewer's suggestion, we included the specific tracts with corresponding references to studies that found evidence of late myelination and early decline of those same brain areas. Discussion (6th paragraph):

" Despite the cross-sectional nature of our study, our findings fit into the hypothesis of antero-posterior spread of pathology⁴², but also follow the postulated last-in-first-out hypothesis, showing an increased vulnerability of late myelinating prefrontal brain regions^{29,43-45} including the cingulum, uncinata fasciculus and superior longitudinal fasciculus which were shown to reach peak myelination later in life^{44,46,47}."

Reviewers' comments:

Reviewer #1 (Remarks to the Author):

The authors responded to most of my concerns. I still have a minor comment to consider: I would suggest to discuss the lack of investigation of the thalamic, striatal and cerebellar tracks as a limitation and mention it as a possible outlook for future studies.

Reviewer #3 (Remarks to the Author):

The manuscript is improved with revisions. However, the absence of data to suggest the clinical meaningfulness of the associations will limit the value of the work for the community. One way of strengthening the work might be to report the magnitude of the relatively large diabetes beta when confounds of HTN, high BMI and dyslipidemia are controlled.

Revised version of the manuscript **COMMSBIO-22-3543-A**, “*Neurobiology of associations between cardiovascular risk factors and anatomy of the ageing brain*” by Trofimova et al.

Reviewers' comments:

Reviewer #1 (Remarks to the Author):

The authors responded to most of my concerns. I still have a minor comment to consider: I would suggest to discuss the lack of investigation of the thalamic, striatal and cerebellar tracks as a limitation and mention it as a possible outlook for future studies.

We thank the Reviewer for pointing out this important limitation. We amended the limitations paragraph in the Discussion (penultimate paragraph):

We acknowledge several limitations of our study. The cross-sectional nature of the study does not allow causal inference about the observed associations. The study sample comes from an urban Swiss population; thus, results might not generalise outside high income countries with predominant Caucasian ethnicity. The relatively large effects reported for diabetes should be interpreted with care given the sample imbalance (6.5% of participants had diabetes) and the high uncertainty of estimates, especially when adjusting for the other CVRFs. **Given the focus on cortico-cortical and cortico-spinal main connections, our analysis did not include striatal, thalamic and cerebellar white matter tracts, which will be included in future studies.** Finally, we used tract-averaged brain metrics, so the spatial distribution of effects within tracts remains unknown.

Reviewer #3 (Remarks to the Author):

The manuscript is improved with revisions. However, the absence of data to suggest the clinical meaningfulness of the associations will limit the value of the work for the community. One way of strengthening the work might be to report the magnitude of the relatively large diabetes beta when confounds of HTN, high BMI and dyslipidemia are controlled.

We thank the Reviewer for their suggestion which we have followed and found that, when controlled for the other 5 risk factors (hypertension, dyslipidemia, smoking, high BMI and WHR),

diabetes associations with higher water content (ISOVF) became weaker, suggesting shared variance with hypertension. Associations with lower axonal density (ICVF) remained of similar effect size but were no longer significant, indicating larger errors and thus, higher uncertainty of results. Overall, the diabetes results of our study should be interpreted with caution, as it was the only risk factor with high sample imbalance - 6.5% of participants had diabetes, whereas the other risk factors included >35% of cases. We have amended the manuscript accordingly and added a Supplementary Figure and detailed results in Supplementary Table 3.

Methods (penultimate paragraph):

As a post-hoc analysis to the relatively large effect sizes observed in diabetes associations, we investigated those associations in models adjusted for all other CVRFs:

$$\begin{aligned} \text{Tract value} = & \beta_0 + \beta_1 \text{diabetes} + \beta_2 \text{hypertension} + \beta_3 \text{highBMI} + \beta_4 \text{highWHR} & 9 \\ & + \beta_5 \text{dyslip} + \beta_6 \text{smoking} + \beta_7 \text{age} + \beta_8 \text{age}^2 + \beta_9 \text{sex} + \beta_{10} \text{TIV} + \varepsilon \end{aligned}$$

where β_1 was the primary estimate of interest, but β_{2-6} are also reported for comparison.

Results (last paragraph of *Cardiovascular risk factors and white matter microstructure*):

The relatively larger effect sizes observed for diabetes and decreased axonal density remained similar when adjusting for the other five CVRFs, but they were no longer significant due to larger standard errors, indicating higher uncertainty of these associations (see Extended Data Figure 10 and Table S3). On the other hand, diabetes and increased water content effect sizes were lower when adjusting for other CVRFs, which could be due to shared variance with hypertension.

Discussion (penultimate paragraph):

The relatively large effects reported for diabetes should be interpreted with caution given the sample imbalance (6.5% of participants had diabetes) and the high uncertainty of estimates, especially when adjusting for the other CVRFs.